# DNMT3A-mediated epigenetic silencing of SOX17 contributes to endothelial cell migration and fibroblast activation in wound healing

Xiaoping Yu[1], Xiaoting Ma[2], Junli Zhou[1]*

1 The Department of Burn, Gansu Provincial Hospital, Lanzhou, China, 2 Gansu University of Chinese Medicine, Lanzhou, China

* zhoujunli11@sohu.com

## Abstract

### Background

Wound healing, especially impaired chronic wound healing, poses a tremendous challenge for modern medicine. Understanding the molecular mechanisms underlying wound healing is essential to the development of novel therapeutic strategies.

### Methods

A wound-healing mouse model was established to analyze histopathological alterations during wound healing, and the expression of SRY-box transcription factor 17 (SOX17), DNA methyltransferase 3 alpha (DNMT3A), and a specific fibroblast marker S100 calcium-binding protein A4 (S100A4) in wound skin tissues was tested by immunofluorescence (IF) assay. Cell proliferation and migration were evaluated using 5-ethynyl-2′-deoxyuridine (EdU) and Transwell migration assays. RT-qPCR and western blotting were used to measure RNA and protein expression. Enzyme-linked immunosorbent assay (ELISA) was performed to detect the secretion of transforming growth factor-beta (TGF-β). Chromatin immunoprecipitation followed by qPCR (ChIP-qPCR) and DNA pull-down assays were performed to confirm the interaction between DNMT3A and the CpG island of the SOX17 promoter. Promoter methylation was examined by pyrosequencing.

### Results

SOX17 and DNMT3A expression were regularly regulated during the different phases of wound healing. SOX17 knockdown promoted HUVEC migration and the production and release of TGF-β. Through establishing an endothelial cells-fibroblasts co-culture model, we found that SOX17 knockdown in HUVECs activated HFF-1 fibroblasts, which expressed α-smooth muscle actin (α-SMA) and type I collagen (COL1). DNMT3A overexpression reduces SOX17 mRNA levels. ChIP-qPCR and DNA pull-down assays verified the interaction between DNMT3A and CpG island in the SOX17 promoter region. Pyrosequencing confirmed that DNMT3A overexpression increased the methylation level of the SOX17 promoter.

**Data Availability Statement:** All relevant data are within the paper and its Supporting information files.

**Funding:** This work was supported by grants from Science and Technology Program of Gansu Province (No. 21JR7RA609). The funders had no role in study design, data collection and analysis, decision to publish, or preparation of the manuscript.

**Competing interests:** The authors declare that they have no conflict of interest.

## Conclusion

DNMT3A-mediated downregulation of SOX17 facilitates wound healing by promoting endothelial cell migration and fibroblast activation.

## Introduction

The human skin acts as a barrier for the human body from ultraviolet light, toxic substances, and pathogenic microorganisms. Wound-healing dysfunction of skin tissues is a serious health burden in many people, including the elderly individuals [1], patients with diabetes [2], and patients receiving chemotherapy or radiotherapy [3]. Compared to the general population, these individuals are more likely to develop painful, non-healing ulcers, which can lead to infections, amputations, and tumor progression.

Wound healing is a complex multistep process involving inflammatory responses, granulation, re-epithelialization, angiogenesis, and tissue remodeling. As the main effector cells in the process of wound healing, the activation of fibroblasts results in the differentiation of myofibroblasts featured by α-smooth muscle actin (α-SMA) expression, which provides cells with contractile activity and promotes wound contraction. Numerous factors regulate the activation of fibroblasts, including cytokines and growth factors [4]. Among these factors, transforming growth factor-beta (TGF-β) is a potent fibrogenic factor [5] produced by various cell types, including vascular endothelial cells [6]. We explored the molecular mechanism behind endothelial cells-derived TGF-β-induced activation of fibroblasts during wound healing.

SRY-box transcription factor 17 (SOX17) belongs to the SOX transcription factor family and plays a vital role in embryonic development and cell fate determination. Angiogenesis is an important process during wound healing. Several studies have reported the pivotal roles of SOX17 and its paralog, SOX7, in development and tumor angiogenesis [7], and SOX17 has been reported to display an endothelial cell-specific expression pattern [8]. However, the physiological role of SOX17 in endothelial cells during wound healing remains unclear.

In this study, we established a wound healing mouse model and found that SOX17 expression was regularly regulated during different phases of wound healing. To analyze the role of SOX17 during wound healing *in vitro*, first, we assessed the effect of SOX17 interference on TGF-β secretion and migration ability in vascular endothelial cells. An endothelial cells-fibroblasts co-culture model *in vitro* was established to dissect the interplay between endothelial cells and fibroblasts upon SOX17 silencing. Finally, we investigated the upstream regulatory mechanisms of SOX17 during wound healing.

## Materials and methods

### Establishment of mice wound healing model

BALB/c mice (7-week-old) were used to establish a wound healing model after 1 week of acclimation. Specifically, hair at the surgical sites were removed the day before the surgery. On the day of the surgery, all mice were anesthetized with isoflurane (Jiupai Company, Hebei, China), and the skin tissues were fully removed to create a wound. Skin wounds were imaged on days 0, 1, 3, 7, and 14, and wound areas were calculated. On days 1, 3, 7, and 14, the mice were sacrificed by $CO_2$ asphyxiation and death was confirmed by the absence of respiration or heartbeat. The injured skin of the mice was lifted with surgical forceps and 2-mm tissues surrounding the injured site were collected. Wound skin tissues were subjected to hematoxylin and eosin (HE),

Masson trichrome, and IF staining. All animal experiments were approved by the Laboratory Animal Committee of Gansu Provincial Hospital.

## HE staining assay

HE staining was used to analyze tissue morphology and histopathological alterations of the wounded skin tissues. The skin tissues were immersed in 10% neutral formalin solution (Solarbio, Beijing, China) for 72 h for tissue fixation. The tissues were embedded in paraffin wax and cut into pieces. These tissue pieces were stained with HE (Solarbio), and pathological alterations were observed using an optical light microscope (Nikon Eclipse E100, Nikon, Japan).

## Masson trichrome staining

A Masson staining assay was conducted to detect collagen fibers and observe tissue structure and angiogenesis in the wounded skin tissues. Masson's trichrome staining kit (Solarbio) was used in this assay, and the procedure was performed in strict accordance with the manufacturer's instructions.

## IF assay

An IF assay was performed to analyze protein expression in skin tissues or HFF-1 cells according to standard procedures. The expression of S100 calcium-binding protein A4 (S100A4), SOX17, and DNA methyltransferase 3 alpha (DNMT3A) in wounded skin tissues was measured using an IF assay. The skin tissues were fixed, permeabilized, blocked, and labeled with the following antibodies: anti-S100A4 (Cat No. 16105-1-AP; 1:300; Proteintech, Wuhan, China), anti-SOX17 (Cat No. 24903-1-AP; 1:300; Proteintech), and anti-DNMT3A (Cat No. 20954-1-AP; 1:500; Proteintech). DAPI (Cat No. G1012; Servicebio, Wuhan, China) was used to mark cell nucleus. The fluorescence intensity was observed under a confocal microscope at ×200 magnification.

The levels of α-SMA and type I collagen (COL1) in HFF-1 cells were also detected by IF assay using the following antibodies: anti-α-SMA (Cat No. GB111364; 1:500; Servicebio) and anti-COL1 (Cat No. GB11022; 1:300; Servicebio).

## Cell culture

Human umbilical vein endothelial cells (HUVECs) were purchased from Procell Biotechnology (Wuhan, China) and were maintained in endothelial cell medium (ECM) (ScienCell, Carlsbad, CA, USA) containing 5% FBS (Thermo Scientific, Rockford, IL, USA) in a humidified incubator ($37^\circ$C, 5% $CO_2$).

Human foreskin fibroblast cell line (HFF-1) was purchased from Procell Biotechnology and was maintained in DMEM (Procell Biotechnology) plus 10% FBS (Thermo Scientific) in a humidified incubator ($37^\circ$C, 5% $CO_2$).

## Cell transfection

Three short hairpin RNAs (shRNAs) targeting SOX17 (sh-SOX17#1, #2, and #3) and their negative control empty vector pLKO.1-copGFP-PURO (sh-NC) were purchased from TsingKe Biotechnology (Beijing, China). Following are the target sequences of sh-SOX17#1, #2, and #3:

Sh-SOX17#1: 5'-GCACGGAATTTGAACAGTA-3'.

Sh-SOX17#2: 5'-GGTATATTACTGCAACTAT-3'.

Sh-SOX17#3: 5'-GGGCTAAGGACGAGCGCAA-3'.

The DNMT3A overexpression plasmid (pCMV-DNMT3A(human)-Neo) was purchased from Miaolingbio (Wuhan, China), and an empty pCMV vector was used as a control.

Lipofectamine 3000 reagent (Invitrogen, Carlsbad, CA, USA) was utilized for transient transfection.

### 5-ethynyl-2′-deoxyuridine (EdU) assay

EdU assay was conducted using a Click-iT EdU-555 cell proliferation detection kit (Servicebio). After transfection for 48 h, HUVECs were incubated with EdU reagent for 2 h. After fixing with 4% paraformaldehyde for 30 min and incubating with 1×Apollo staining solution for 30 min, cells were mixed with 0.5% TritonX-100 for 10 min. Cell nucleus was marked with Hoechst 33342 for 30 min at room temperature. The rate of positive cells (red dots/blue dots) was analyzed using a fluorescence confocal microscope.

### Transwell migration assay

Transwell chambers (Corning Costar Corporation, Corning, NY, USA) were used to analyze cell migration. HUVECs were transfected with sh-NC or sh-SOX17 for 48 h and then reseeded in the upper transwell chambers in serum-free medium. The lower chambers were filled with the culture medium supplemented with 10% FBS. Cells were incubated for an additional 24 h at 37°C. The non-migrated cells on the top side of the transwell filter were removed using a cotton swab, whereas the migrated cells on the lower surface of the transwell filter were immobilized with 4% paraformaldehyde (Beyotime) for 30 min and then stained with 0.1% crystal violet solution (Beyotime) for 10 min. Cells that migrated to five random fields were counted under an optical light microscope.

### RT-qPCR

Total RNA was extracted from HUVECs using TRIzol reagent (Beyotime). cDNA was obtained using a PrimeScript RT Reagent Kit (Takara, Dalian, China). qPCR was performed using the SYBR Green qRT-PCR kit (Takara) on an ABI viiA7 Real-Time PCR System (Applied Biosystems, Foster City, CA, USA). GAPDH was used as an internal control. The fold changes were analyzed by relative quantification ($2^{-\Delta\Delta Ct}$). The specific primers of SOX17 and TGF-β were listed in S1 Table.

### Western blot assay

Total cell lysates were prepared using RIPA lysis buffer (Beyotime) supplemented with a protease inhibitor cocktail (Bio-Rad, Hercules, CA, USA). Protein concentration was measured using a commercial bicinchoninic acid (BCA) Protein Assay Kit (Beyotime). Protein samples (30 μg) were separated using sodium dodecyl sulfate-polyacrylamide gel electrophoresis and transferred onto PVDF membranes (Millipore, Billerica, MA, USA). The non-specific sites on the PVDF membranes were sealed by incubating with 5% non-fat milk for 1 h. Subsequently, the PVDF membranes were labeled with the primary antibodies as indicated below: anti-TGF-β (Cat No. 21898-1-AP; 1:3000; Proteintech) and anti-DNMT3A (Cat No. 20954-1-AP; 1:10000; Proteintech). Thereafter, the PVDF membranes were labeled with a horseradish peroxidase-linked secondary antibody for 2 h at room temperature. Immunoblots were visualized using an ECL kit (Beyotime), and ImageJ software (NIH, Bethesda, MD, USA) was used for protein quantification.

## Detection of TGF-β concentration

The concentration of TGF-β in the culture supernatant was evaluated using a Human TGF-β1 ELISA Kit (Cat No. EK981; Multi Sciences; Hangzhou, China).

## Co-culture of HUVECs and HFF-1 cells

After transfection for 48 h, the culture supernatant of HUVECs was collected and culture HFF-1 cells for 24 h.

## Cell counting kit-8 (CCK8) assay

Cell viability was measured by CCK8 assay using a CCK8 kit (Beyotime, Shanghai, China). After the specific treatment, cells in 96-well plates were incubated in the presence of 10 μL CCK8 reagent/well for 4 h at 37˚C. The absorbance was measured at 450 nm using a microplate reader (K3, Thermo Scientific).

## Chromatin immunoprecipitation followed by PCR or qPCR (ChIP-PCR or ChIP-qPCR)

The binding relationship between DNMT3A and the CpG island of the SOX17 promoter was tested using the BeyoChIP$^{TM}$ Enzymatic Chromatin Immunoprecipitation Assay Kit (Beyotime). Briefly, DNA was fragmented into 150–1000 bp fragments using an ultrasonic cell-crushing device (Scientz Biotechnology, Ningbo, China). Subsequently, the protein-DNA complex was mixed with the antibody of IgG or DNMT3A at 4˚C overnight. DNA samples were purified from the precipitates after cross-link reversal and subjected to PCR or qPCR using primers specific for the SOX17 promoter CpG island. Primers used are listed in S1 Table.

## DNA pull-down assay

The binding relationship between DNMT3A and the SOX17 promoter was tested using the Pure Magnetic DNA-Protein Pull down Kit (Writegene, Henan, China). Briefly, a biotin-labeled SOX17 promoter probe (Bio-SOX17-pro) and a non-labeled promoter probe (SOX17--pro) were synthesized. The probes were mixed with beads and incubated overnight with total cellular protein lysates. Finally, the DNA-protein complex was pulled down by the beads, and the enrichment of DNMT3A protein was evaluated by western blotting.

## DNA methylation analysis

Pyrosequencing was performed to analyze DNA methylation. After transfection, genomic DNA was extracted using a genomic DNA extraction kit (Qiagen, Hilden, Germany) and the DNA samples were processed using a Qiagen EpiTect Bisulfite Kit (Qiagen). Following are the PCR amplification system (50 μL) requirements: 34.8 μL $H_2O$, 10 μL 5×buffer GC (KAPA), 1 μL dNTP (10 mM/each), 1 μL primer (up 50 pM/μL), 1 μL primer (down 50 pM/μL), 2 μL template, and 0.2 μL Taq (5 U/μL). The primers were synthesized by the Shenzhen Huada Gene Research Institute (Shenzhen, China) and the primer sequences are listed in S1 Table. The samples were analyzed on a PyroMark Q48 platform, and the methylation status of each site was analyzed using Pyro Q-CpG software of the pyrophosphate sequencer.

## Data analysis

All data were analyzed by GraphPad Prism 8.0 software (GraphPad Prism, La Jolla, CA, USA) and were represented as the form of mean ± SD. Unless otherwise stated, all experiments were performed in triplicates. Differences were analyzed using Student's $t$ test or one-way ANOVA, followed by Dunnett's test. $P<0.05$ was considered to indicate statistically significant differences.

## Results

### Construction of wound healing model in mice

We developed a wound-healing mouse model to monitor the key processes involved in wound healing on days 0, 1, 3, 7, or 14 after model establishment. The wound area gradually decreased (Fig 1A). Hematoxylin and eosin (HE) and Masson staining were performed to evaluate the histopathological alterations in wounded skin tissues. As shown in Fig 1B, the epidermis of the skin tissues on day 1 was thin, the structure of each layer was clear, the connective tissue of the dermis was neatly arranged, and the appendages of the skin were scattered, such as hair follicles and sweat glands. No subcutaneous tissues and muscle layer were observed, and no obvious inflammatory cell infiltration was observed. On days 3 and 7, local infiltration of inflammatory cells was observed (black arrow); an increase in the number of new blood vessels was observed (red arrow); no skin appendages were observed, such as hair follicles and sweat glands; exfoliation of necrotic tissue was occasionally observed in the skin tissues (blue arrow); and the dermis was repaired by proliferating connective tissue, with numerous fibroblasts and collagen fibers (green arrow). On day 14, the structure of each layer of the skin tissue was clear, and the connective tissue of the dermis was neatly arranged. Similar to the HE staining assay, Masson staining showed that the appendages of the skin were scattered on day 1 and the organization structure was tight (Fig 1C). On day 3, the skin tissue structure was loose and disordered (black arrow), and the lesions were severe. On day 7, a significant number of new blood vessels was observed (red arrows). On day 14, the overall structure and organization of skin tissues were normal. These data suggest the recruitment of inflammatory cells, the activation of fibroblasts, and angiogenesis are central processes during wound healing.

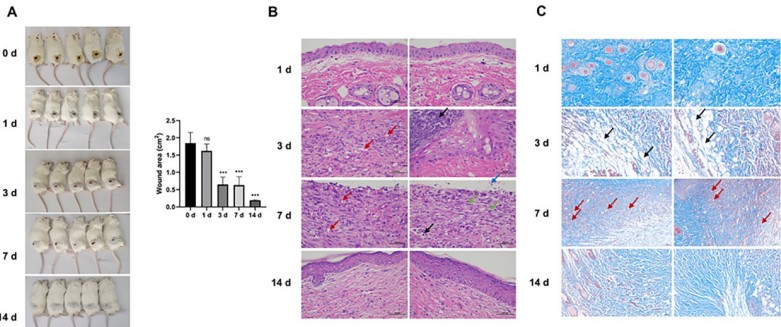

**Fig 1. Construction of wound healing model in mice.** BALB/c mice (7-week-old) were used to establish wound healing model after 1 week of acclimation. (A) The average wound area of mice on day 0, 1, 3, 7, or 14 after model establishment was analyzed. (B and C) The key processes involved in wound healing were analyzed by HE and Masson staining assays. ***$P<0.001$. ns: not statistically significant.

## IF staining of fibroblast marker S100A4 and interested proteins

We determined the expression of a specific fibroblast marker, S100A4, in wounded skin tissue using an IF assay. Fibroblasts play vital roles in the proliferative phase of wound healing [9]. The data revealed that S100A4 was dramatically upregulated on day 7 after model establishment (Fig 2A), suggesting that day 7 of wound healing may be the stage of fibroblast proliferation and collagen deposition. Several studies have reported a pivotal regulatory role of SOX17 in angiogenesis [10–12]. We found that SOX17 levels in wound skin tissues gradually increased from day 1 to day 7, and its expression decreased on day 14 (Fig 2B). DNMT3A can mediate the epigenetic gene silencing through DNA methylation [13, 14]. We found that DNMT3A expression was gradually downregulated as time progressed from day 1 to day 7, and its level was markedly elevated on day 14 (Fig 2C), in contrast to the SOX17 level. These data suggest that SOX17 and DNMT3A are involved in wound healing.

## SOX17 interference promotes the migration and secretion of TGF-β in HUVECs

RT-qPCR and IF assays were conducted to analyze the knockdown efficiencies of the three shRNAs targeting SOX17 (sh-SOX17#1, #2, and #3). As sh-SOX17#2 exhibited the highest knockdown ability among the three shRNAs (Fig 3A and 3B), it was selected for subsequent experiments and named sh-SOX17. We first assessed the regulatory role of SOX17 in the proliferation of HUEVCs through EdU assay. HUVECs were transfected with sh-NC or sh-SOX17 for 48 h. The data revealed that SOX17 knockdown had a slight but not significant effect on the proliferation of HUVECs (Fig 3C). The migration of vascular endothelial cells plays a key role in the generation of new blood vessels and vessel remodeling during wound healing [15]. The data of transwell migration assay suggested that migrated cell number was notably increased in SOX17-silenced group relative to sh-NC group (Fig 3D), suggesting that SOX17 interference facilitated HUVEC cell migration. Wound healing is accompanied by an interplay between multiple cell types such as vascular endothelial cells, fibroblasts, neutrophils, and macrophages. TGF-β is a well-known vital regulator on fibroblasts activation and fibrosis, and it is released by vascular endothelial cells, neutrophils, keratinocytes, and macrophages [6, 16]. Here, we analyzed the SOX17 role on TGF-β-mediated interaction between vascular endothelial cells and fibroblasts. RT-qPCR and western blot assays together demonstrated that SOX17 silencing enhanced the mRNA and protein abundance of TGF-β in HUVECs (Fig 3E

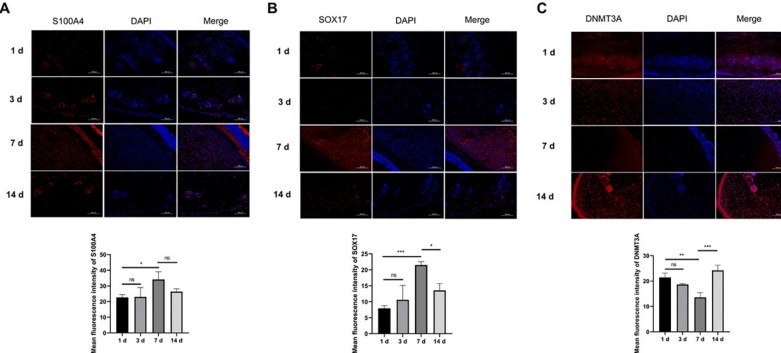

**Fig 2. IF staining of fibroblast marker S100A4 and interested proteins.** (A-C) IF assay was performed to evaluate the abundance of fibroblast markers (S100A4) and proteins of interest (SOX17 and DNMT3A) in wounded skin tissues. *$P<0.05$, **$P<0.01$, ***$P<0.001$. ns: not statistically significant.

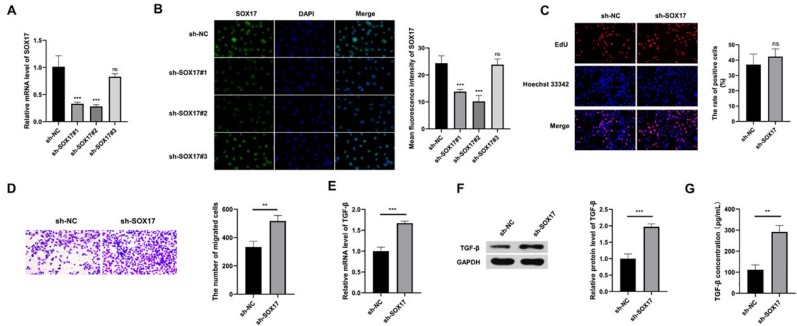

**Fig 3. SOX17 interference promotes the migration and the secretion of TGF-β in HUVECs.** (A and B) The knockdown efficiencies of three shRNAs targeting SOX17 in HUVECs were analyzed by RT-qPCR and IF assays. Sh-SOX17#2 with the highest knockdown efficiency among three shRNAs was named as sh-SOX17 in the subsequent experiments. (C-G) HUVECs were transfected with sh-NC or sh-SOX17 for 48 h. (C) Cell proliferation was assessed by EdU assay. (D) Transwell migration assay was implemented to evaluate the migration ability of transfected HUVECs. (E and F) Intracellular TGF-β level was determined through RT-qPCR and western blot assay. (G) ELISA was implemented to analyze the secretion of TGF-β in the culture supernatant. **$P<0.01$, ***$P<0.001$. ns: not statistically significant.

and 3F). According to ELISA, SOX17 knockdown promoted TGF-β secretion in the culture supernatant of HUVECs (Fig 3G). In the subsequent experiments, we further assessed the effects of SOX17 knockdown-mediated TGF-β secretion in HUVECs on the activation of fibroblasts.

## SOX17 silencing activates HFF-1 fibroblasts

After transfection for 48 h, the culture supernatant of HUVECs was collected and HFF-1 cells were cultured for 24 h. SOX17 knockdown increased HFF-1 cell viability (Fig 4A). The

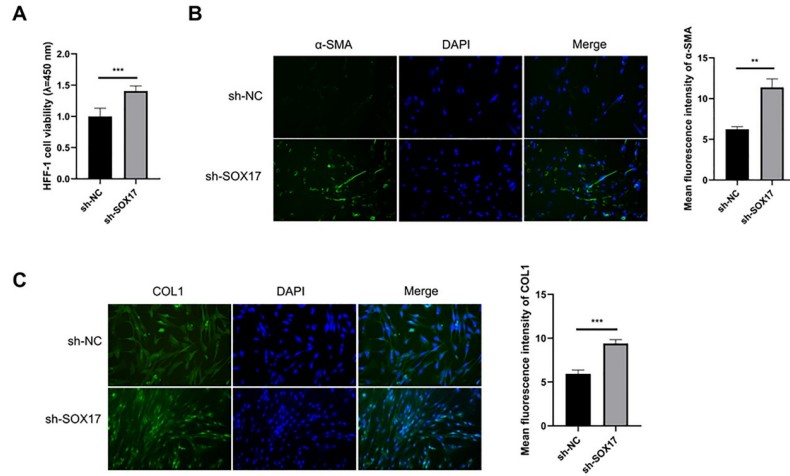

**Fig 4. SOX17 silencing elevates the viability and the expression of two markers of fibroblast activation.** (A-C) HUVECs were introduced with sh-NC or sh-SOX17 for 48 h, and the culture supernatant was collected to culture HFF-1 cells for 24 h. (A) Cell viability of HFF-1 cells was analyzed by CCK8 assay. (B and C) IF assay was employed to analyze the abundance of two markers of fibroblast activation (α-SMA and COL1) in HFF-1 cells. **$P<0.01$, ***$P<0.001$.

important hallmarks of fibroblast activation are the expression of α-SMA and the deposition of collagen [17]. The data from the IF assay revealed that SOX17 silenced-HUVECs-derived conditioned medium significantly enhanced the abundance of α-SMA and COL1 (Fig 4B and 4C). In summary, SOX17 interference in HUVECs promoted the activation of HFF-1, and this effect might be mediated by the increased production of TGF-β.

## DNMT3A negatively regulates SOX17 expression by increasing its DNA methylation

The human genome contains three DNA methyltransferases with catalytic activity, namely DNMT1, DNMT3A, and DNMT3B [18]. DNA methylation is a covalent modification that occurs at the CpG dinucleotides of target genes and can mediate the epigenetic silencing of gene expression. Considering the opposite expression patterns of SOX17 and DNMT3A during wound healing (Fig 2B and 2C), we hypothesized that the DNA methyltransferase DNMT3A negatively regulates SOX17 expression by increasing the methylation level of its promoter. RT-qPCR showed that DNMT3A overexpression reduced the mRNA level of SOX17 in HUVECs (Fig 5A). We designed specific primers for the SOX17 promoter CpG island and explored the interaction between DNMT3A and SOX17 promoter CpG islands using ChIP-PCR and ChIP-qPCR assays. A specific band of approximately 250 bp was observed in the input and anti-DNMT3A groups (Fig 5B), verifying the binding between DNMT3A and SOX17 promoter CpG islands. The ChIP-qPCR assay showed a significant enrichment of the SOX17 promoter CpG island in the anti-DNMT3A group compared with the anti-IgG group (Fig 5C). In turn, a DNA pull-down assay with a biotin-labeled SOX17 promoter probe was conducted, and the results revealed a marked enrichment of DNMT3A in the Input and Bio-SOX17-pro groups (Fig 5D), further confirming that DNMT3A could bind to the CpG island of the SOX17 promoter. Subsequently, we explored whether DNMT3A was involved in the differential regulation of SOX17 methylation using pyrosequencing, a method for accurately quantifying DNA methylation at specific CpG sites. Methyl Primer Express Software (v1.0) was used to predict CpG islands in the promoter region of SOX17 and to design pyrosequencing primers. The methylation level of the SOX17 promoter in HUVECs transfected with the vector or DNMT3A plasmid was analyzed. Pyrosequencing results showed that the methylation level of the SOX17 promoter in the DNMT3A group was higher than that in the vector group (54.16% vs. 51.77%; Fig 5E), suggesting that DNMT3A overexpression increased the methylation level of the SOX17 promoter. These data verified that DNMT3A negatively regulated SOX17 expression in HUVECs by increasing its DNA methylation levels.

## Discussion

Wound healing can be restrained and prolonged in several diseases, resulting in chronic non-healing wounds and posing a health burden on many individuals [19]. Understanding the molecular mechanisms underlying the coordination of different cell types during wound healing is essential for shortening the healing time. Although various treatment strategies have been implemented to facilitate wound healing, an optimal approach is still being developed. Wound healing is accompanied by inflammation, granulation, reepithelialization, angiogenesis, and tissue remodeling [20]. In this study, a wound-healing mouse model revealed that the infiltration of inflammatory cells, generation of new blood vessels, and collagen deposition are central processes during wound healing. In addition, SOX17 and DNMT3A expression levels in wounded skin tissues are regularly regulated during wound healing.

SOX17 is a critical regulator of angiogenesis, and its function in angiogenesis may vary under different conditions. Kim *et al.* found that SOX17 inhibits angiogenesis in a high-grade

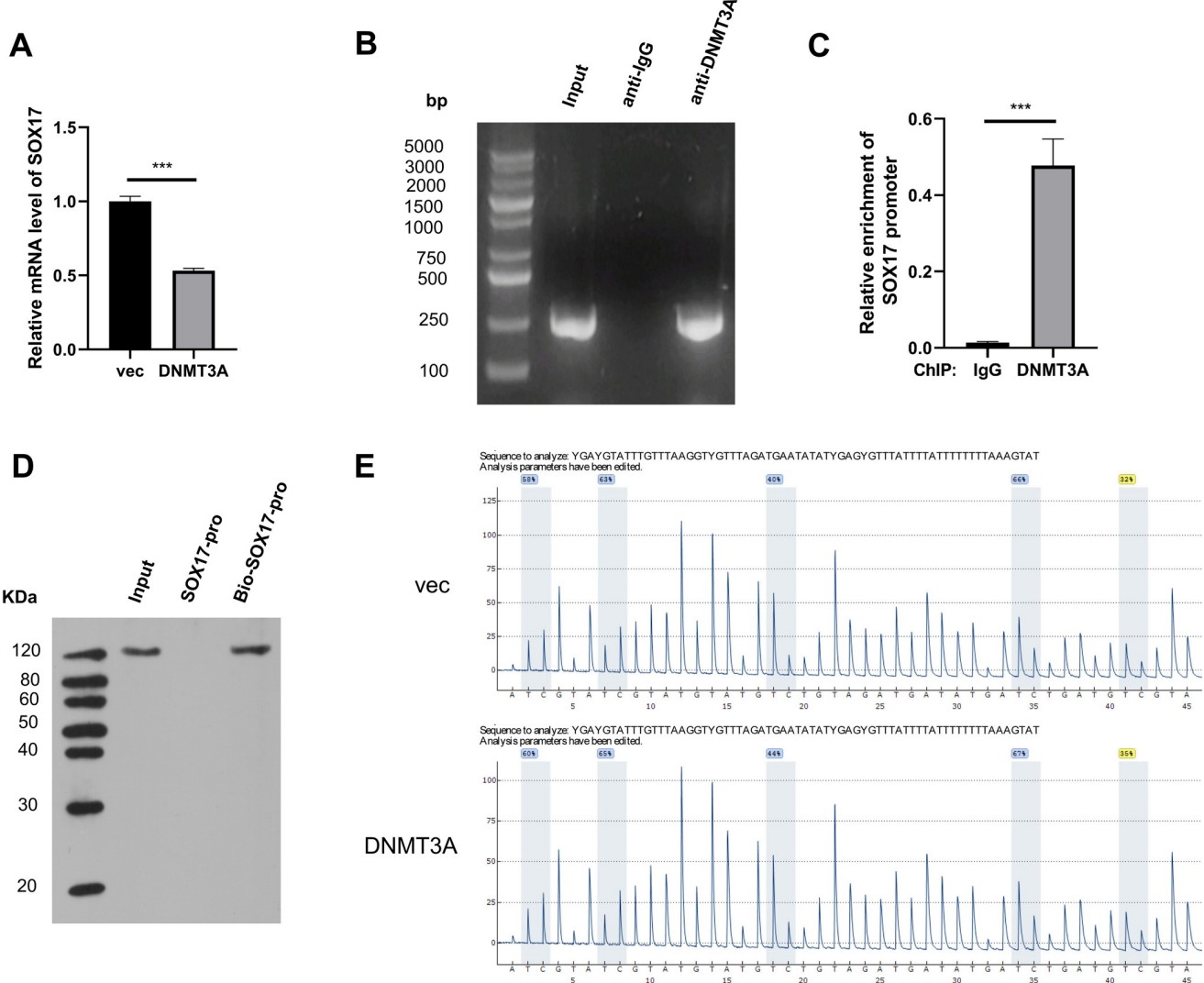

**Fig 5. DNMT3A negatively modulates SOX17 expression by increasing its DNA methylation.** (A) HUVECs were transfected with vector or DNMT3A overexpression plasmid, and SOX17 protein abundance was detected via western blot assay. (B and C) ChIP-PCR and ChIP-qPCR were conducted to verify the interaction between DNMT3A and SOX17 promoter. (D) DNA pull-down assay using a biotin-labeled SOX17 promoter probe was performed to confirm the binding relation between DNMT3A and SOX17 promoter. (E) Pyrosequencing analysis was conducted to quantify the DNA methylation of SOX17 promoter upon the transfection of vector or DNMT3A overexpression plasmid. ***P<0.001.

glioma model [21]. Another study demonstrated that PAX8 and SOX17 facilitated the secretion of angiogenic factors by ovarian cancer cells [10]. Migration of endothelial cells is vital for various biological processes such as angiogenesis and vessel remodeling [22, 23]. To explore the role of SOX17 in angiogenesis during wound healing, we assessed its effect of SOX17 on endothelial cell migration. SOX17 knockdown markedly promotes endothelial cell migration. Multiple cytokines are involved in the regulation of wound healing process, including FGF, VEGF, TGF-β, and EGF [24]. Among these factors, TGF-β is a critical factor secreted by endothelial cells and other cells, and it regulates fibroblast proliferation and collagen synthesis [5]. We found that SOX17 knockdown promoted the production and release of TGF-β in HUVECs.

The wound healing process involves the interplay between multiple cell types such as endothelial cells, fibroblasts, keratinocytes, neutrophils, and macrophages [25]. Here, we explored whether SOX17 silencing affected the interplay between endothelial cells and fibroblasts through regulating TGF-β secretion, and the endothelial cells-fibroblasts co-culture model *in vitro* was established. These data revealed that SOX17 knockdown increased the viability of HFF-1 cells. In addition, SOX17 knockdown facilitated the activation of HFF-1 cells, as evidenced by the increased expression of fibroblast activation markers, including α-SMA and COL1. These data verified that SOX17 knockdown in HUVECs promoted the viability and activation of HFF-1 cells, and these effects might be mediated by increased release of TGF-β.

DNMT3A is a member of the DNA methyltransferase family and plays a vital role in de novo methylation [26]. DNA methylation is an important epigenetic mechanism that negatively regulates gene expression by modifying CpG sites in the promoter region [27]. The RT-qPCR assay demonstrated that SOX17 mRNA abundance was negatively modulated by DNMT3A in HUVECs. In addition, ChIP-qPCR and DNA pull-down assays verified the interaction between DNMT3A and CpG sites in the SOX17 promoter region. Pyrosequencing analysis showed that DNMT3A enhanced SOX17 promoter methylation. These data indicate that DNMT3A negatively regulates SOX17 expression through DNA methylation.

This study has some limitations that should be considered. Whether the effect of SOX17 knockdown on the viability and activation of HFF-1 cells is mediated by TGF-β needs to be further explored. However, a gene-depleted animal model needs to be established to confirm this conclusion.

In conclusion, our data showed that DNMT3A-mediated SOX17 silencing promotes wound healing by inducing the migration of endothelial cells and activating fibroblasts. Thus, DNMT3A and SOX17 are potential novel targets for enhancing wound healing.

## Supporting information

**S1 Table. Primer sequences used in this study.**
(DOCX)

**S1 File. The original uncropped and unadjusted images underlying all blot or gel results.**
(DOCX)

**S2 File.**
(DOCX)

**S1 Graphical abstract.**
(TIF)

## Acknowledgments

I express my gratitude to all those who helped me write this thesis. I would also like to thank Mr. Xiaoping Yu and Ms. Xiaoting Ma, who contributed to this research.

## Author Contributions

**Data curation:** Xiaoping Yu, Xiaoting Ma.

**Writing – original draft:** Xiaoping Yu.

**Writing – review & editing:** Junli Zhou.

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
