## [Decision Letter · Decision Letter 0]

15 Aug 2023

PONE-D-23-16896DNMT3A-mediated epigenetic silencing of SOX17 contributes to endothelial cell migration and fibroblast activation during wound healingPLOS ONE

Dear Dr. Zhou,

Thank you for submitting your manuscript to PLOS ONE. After careful consideration, we feel that it has merit but does not fully meet PLOS ONE’s publication criteria as it currently stands. Therefore, we invite you to submit a revised version of the manuscript that addresses the points raised during the review process.

We look forward to receiving your revised manuscript.

Kind regards,

Peng Zhang, Ph.D.

Academic Editor

PLOS ONE

2. To comply with PLOS ONE submissions requirements, in your Methods section, please provide additional information regarding the experiments involving animals and ensure you have included details on (1) methods of sacrifice, (2) methods of anesthesia and/or analgesia, and (3) efforts to alleviate suffering

A clean copy of the edited manuscript (uploaded as the new *manuscript* file).

“This work was supported by grants from Science and Technology Program of Gansu Province (No. 21JR7RA609).”

“I would like to express my gratitude to all those who have helped me during the writing of this thesis. I gratefully acknowledge the help of Science and Technology Program of Gansu Province (No. 21JR7RA609) that funded this research. Also, I would like to thank Mr. Xiaoping Yu, Ms. Xiaoting Ma, who contributed to the research work.”

“This work was supported by grants from Science and Technology Program of Gansu Province (No. 21JR7RA609).”

7. PLOS requires an ORCID iD for the corresponding author in Editorial Manager on papers submitted after December 6th, 2016. Please ensure that you have an ORCID iD and that it is validated in Editorial Manager. To do this, go to ‘Update my Information’ (in the upper left-hand corner of the main menu), and click on the Fetch/Validate link next to the ORCID field. This will take you to the ORCID site and allow you to create a new iD or authenticate a pre-existing iD in Editorial Manager. Please see the following video for instructions on linking an ORCID iD to your Editorial Manager account: https://www.youtube.com/watch?v=_xcclfuvtxQ.

8. Your ethics statement should only appear in the Methods section of your manuscript. If your ethics statement is written in any section besides the Methods, please move it to the Methods section and delete it from any other section. Please ensure that your ethics statement is included in your manuscript, as the ethics statement entered into the online submission form will not be published alongside your manuscript.

Reviewers' comments:

Reviewer's Responses to Questions

**Comments to the Author**

1. Is the manuscript technically sound, and do the data support the conclusions?

Reviewer #1: Yes

Reviewer #2: Partly

2. Has the statistical analysis been performed appropriately and rigorously? 

Reviewer #1: Yes

Reviewer #2: Yes

3. Have the authors made all data underlying the findings in their manuscript fully available?

Reviewer #1: Yes

Reviewer #2: Yes

4. Is the manuscript presented in an intelligible fashion and written in standard English?

Reviewer #1: Yes

Reviewer #2: Yes

5. Review Comments to the Author

Reviewer #1: 1.Please include complete information about reagents in Methods

2.Most of the references are not latest, please update your references which published within three years

3.All the figures' quality is so poor, please revise it.

4.Organize all primer sequences in one table.

Reviewer #2: The authors demonstrate that DNMT3A-mediated down-regulation of SOX17 promotes endothelial cell migration and TGF-β secretion, which activate fibroblasts and facilitate wound healing. The authors show that SOX17 and DNMT3A have opposite expressions during the process of wound healing. SOX17 knockdown can promote HUVEC cell migration and enhance production of TGF-β. SOX17 silenced-HUVECs-derived conditioned medium co-culture with HFF-1 cells can promote HFF-1 activation. DNMT3A negatively regulated SOX17 expression in HUVECs by increasing its DNA methylation level. There are several issues that I hope the authors could clarify.

Major

1. The in vivo study shows a significant number of new blood vessel are observed on day 7 in wound skin tissue. In vitro study, SOX17 knockdown facilitate HUVEC cell migration (Fig.3D). However, SOX17 level in wound skin tissues is increased from day 1 to day 7 (Fig.2B). The in vivo and in virto results are reversed. The results are confusing.

2. Transfection of sh-SOX17 for 48h can suppress HUVEC cell viability (Fig.3C) and the authors select 48h transfection for follow-up experiments. Moreover, the following result shows that HUVEC migration number is increased after sh-SOX17 transfection 48h (Fig.3D). How could a decrease in cell viability lead to an increase in migration? The results are puzzling, and the authors need to explain the reasons for the choice of sh-SOX17 transfection time.

3. The authors use the culture medium of cultured HUVEC cells to co-culture with HFF-1 cells. Since ECM medium is used in HUVEC cells culture, while HFF-1 cells use DMEM medium, these two mediums contain different media components, ECM medium contains a variety of growth factors, whether it has an impact on the growth of HFF-1 cells remains to be determined. In addition, HUVEC cell medium suspension contained a variety of HUVEC secreted factors, and it is not clear that TGF-β in the suspension activated HFF-1 fibroblasts. Blocking TGF-β in HUVEC medium suspension and co-culture with HFF-1 can more effectively prove that TGF-β in medium suspension can activate HFF1 fibroblasts.

4. The immunofluorescence staining of the S100A4, SOX17 and DNMT3A are performed on three different sections (Fig.2), and it is better to stain them on the same section.

5. DNMT3A overexpression reduces the mRNA level of SOX17 in HUVECs, and the authors conclude that DNMT3A negative regulation SOX17 promotes HUVECs migration and wound healing. On day 3 and 7, an increase in the number of new blood vessels is observed, combined with the authors' conclusions, this means the expression of DNMT3A is higher elevated and the expression of SOX17 is decreased on day 3 and 7. However, the in vivo study shows DNMT3A expression is down-regulated with the time extending from day 1 to day 7 and SOX17 expression is opposite of DNMT3A (Fig.2). The authors should explain the these contrary results.

Minor

1. There are some inaccuracies in the use of words. For example, in the second paragraph of the discussion section, the authors state that “SOX17 interference induced endothelial cell migration”, among them, “induced” is used inaccurately. In addition, in the third paragraph, “absence” is an inaccurate use in “These data verified that SOX17 absence in HUVECs promoted the viability and activation of HFF-1 cells”.

6. PLOS authors have the option to publish the peer review history of their article (what does this mean?). If published, this will include your full peer review and any attached files.

Reviewer #1: No

Reviewer #2: No

---

## [Author Response · Author response to Decision Letter 0]

16 Sep 2023

Re: Thank you for your kind reminder. We have carefully revised the manuscript and named the files to meet PLOS ONE’s style requirements.

Re: Thank you for your suggestion. The method of animal study was shown as below and the above details have been supplemented.

Establishment of mice wound healing model

BALB/c mice (7-week-old) were used to establish wound healing model after 1-week acclimation. Specifically, hair in the surgery sites was removed the day before surgery. On the day of surgery, all mice were anaesthetized with isoflurane (Jiupai Company, Hebei, China). At the start of surgery, the mice should be fully relaxed, unresponsive to painful stimuli, and the breathing and heart rate were normal. The skin tissues were fully removed to create the wound. Skin wounds were imaged on day 0, 1, 3, 7, and 14, and the wound areas were calculated. On day 1, 3, 7, and 14, mice were sacrificed by CO2 asphyxiation, and mice death was confirmed by absence of respiration or heartbeat. The injured skin of the mice was lifted with surgical forceps and 2 mm tissues surrounding the injured site were taken. The wound skin tissues were subjected to HE staining, Masson trichrome staining, and IF staining. The procedures of animal experiment were approved by the Laboratory Animal Committee of Gansu Provincial Hospital.

efforts to alleviate suffering: At the start of surgery, the mice should be fully relaxed, unresponsive to painful stimuli, and the breathing and heart rate were normal. Mice death was confirmed by absence of respiration or heartbeat.

A clean copy of the edited manuscript (uploaded as the new *manuscript* file).

Re: Thank you for your suggestion. Our manuscript has been polished by American Journal Experts (AJE), and we have up-loaded the polishing certificate.

Re: We have up-loaded the original uncropped and unadjusted images underlying all blot or gel results in Supporting Information files.

“This work was supported by grants from Science and Technology Program of Gansu Province (No. 21JR7RA609).”

Re: The funders had no role in study design, data collection and analysis, decision to publish, or preparation of the manuscript.

“I would like to express my gratitude to all those who have helped me during the writing of this thesis. I gratefully acknowledge the help of Science and Technology Program of Gansu Province (No. 21JR7RA609) that funded this research. Also, I would like to thank Mr. Xiaoping Yu, Ms. Xiaoting Ma, who contributed to the research work.”

“This work was supported by grants from Science and Technology Program of Gansu Province (No. 21JR7RA609).”

Re: Thank you for your suggestion. We have deleted funding information in the Acknowledgments Section.

7. PLOS requires an ORCID iD for the corresponding author in Editorial Manager on papers submitted after December 6th, 2016. Please ensure that you have an ORCID iD and that it is validated in Editorial Manager. To do this, go to ‘Update my Information’ (in the upper left-hand corner of the main menu), and click on the Fetch/Validate link next to the ORCID field. This will take you to the ORCID site and allow you to create a new iD or authenticate a pre-existing iD in Editorial Manager. Please see the following video for instructions on linking an ORCID iD to your Editorial Manager account: https://www.youtube.com/watch?v=_xcclfuvtxQ.

Re: Thank you for your suggestion. We have validated the ORCID in Editorial Manager.

8. Your ethics statement should only appear in the Methods section of your manuscript. If your ethics statement is written in any section besides the Methods, please move it to the Methods section and delete it from any other section. Please ensure that your ethics statement is included in your manuscript, as the ethics statement entered into the online submission form will not be published alongside your manuscript.

Re: Thank you for your suggestion. We have deleted the ethics statement in “Ethics approval” section, and ethics statement only appears in the Methods section of the revised manuscript now.

Re: We have added “Supporting Information” section at the end of the manuscript.

Reviewers' comments:

Reviewer's Responses to Questions

Comments to the Author

1. Is the manuscript technically sound, and do the data support the conclusions?

Reviewer #1: Yes

Reviewer #2: Partly

2. Has the statistical analysis been performed appropriately and rigorously?

Reviewer #1: Yes

Reviewer #2: Yes

3. Have the authors made all data underlying the findings in their manuscript fully available?

Reviewer #1: Yes

Reviewer #2: Yes

4. Is the manuscript presented in an intelligible fashion and written in standard English?

Reviewer #1: Yes

Reviewer #2: Yes

5. Review Comments to the Author

Reviewer #1: 1.Please include complete information about reagents in Methods

Re: Thanks for your professional review work on our article and your kind suggestion. We have supplemented the manufacturers of the reagents in the methods section, including isoflurane (Jiupai Company, Hebei, China), 10% neutral formalin solution (Solarbio, Beijing, China), hematoxylin and eosin (Solarbio), 5% FBS (Thermo Scientific, Rockford, IL, USA), 4% paraformaldehyde (Beyotime), 0.1% crystal violet solution (Beyotime), and protease inhibitor cocktail (Bio-Rad, Hercules, CA, USA).

The city and country of the manufacturer was indicated at the first appearance at the methods section, and only the name of manufacturer was indicated at the second and subsequent appearances.

2.Most of the references are not latest, please update your references which published within three years.

Re: Thank you for your valuable suggestion. We have updated the references in our revised manuscript. The revised references are listed as below.

References

1. Xia W, Li M, Jiang X, Huang X, Gu S, Ye J, et al. Young fibroblast-derived exosomal microRNA-125b transfers beneficial effects on aged cutaneous wound healing. Journal of nanobiotechnology. 2022;20(1):144. doi:10.1186/s12951-022-01348-2

2. Aitcheson SM, Frentiu FD, Hurn SE, Edwards K, Murray RZ. Skin Wound Healing: Normal Macrophage Function and Macrophage Dysfunction in Diabetic Wounds. Molecules (Basel, Switzerland). 2021;26(16). doi:10.3390/molecules26164917

3. Beyene RT, Derryberry SL, Jr., Barbul A. The Effect of Comorbidities on Wound Healing. The Surgical clinics of North America. 2020;100(4):695-705. doi:10.1016/j.suc.2020.05.002

4. Talbott HE, Mascharak S, Griffin M, Wan DC, Longaker MT. Wound healing, fibroblast heterogeneity, and fibrosis. Cell stem cell. 2022;29(8):1161-80. doi:10.1016/j.stem.2022.07.006

5. Peng D, Fu M, Wang M, Wei Y, Wei X. Targeting TGF-β signal transduction for fibrosis and cancer therapy. Molecular cancer. 2022;21(1):104. doi:10.1186/s12943-022-01569-x

6. Frangogiannis N. Transforming growth factor-β in tissue fibrosis. The Journal of experimental medicine. 2020;217(3):e20190103. doi:10.1084/jem.20190103

7. Grimm D, Bauer J, Wise P, Krüger M, Simonsen U, Wehland M, et al. The role of SOX family members in solid tumours and metastasis. Seminars in cancer biology. 2020;67(Pt 1):122-53. doi:10.1016/j.semcancer.2019.03.004

8. Kim D, Grath A, Lu YW, Chung K, Winkelman M, Schwarz JJ, et al. Sox17 mediates adult arterial endothelial cell adaptation to hemodynamics. Biomaterials. 2023;293:121946. doi:10.1016/j.biomaterials.2022.121946

9. Sato K, Asai TT, Jimi S. Collagen-Derived Di-Peptide, Prolylhydroxyproline (Pro-Hyp): A New Low Molecular Weight Growth-Initiating Factor for Specific Fibroblasts Associated With Wound Healing. Frontiers in cell and developmental biology. 2020;8:548975. doi:10.3389/fcell.2020.548975

10. Chaves-Moreira D, Mitchell MA, Arruza C, Rawat P, Sidoli S, Nameki R, et al. The transcription factor PAX8 promotes angiogenesis in ovarian cancer through interaction with SOX17. Science signaling. 2022;15(728):eabm2496. doi:10.1126/scisignal.abm2496

11. Han M, Zhou B. Sox17 and Coronary Arteriogenesis in Development. Circulation research. 2020;127(11):1381-3. doi:10.1161/circresaha.120.318220

12. Qin S, Liu G, Jin H, Chen X, He J, Xiao J, et al. The Dysregulation of SOX Family Correlates with DNA Methylation and Immune Microenvironment Characteristics to Predict Prognosis in Hepatocellular Carcinoma. Disease markers. 2022;2022:2676114. doi:10.1155/2022/2676114

13. Dura M, Teissandier A, Armand M, Barau J, Lapoujade C, Fouchet P, et al. DNMT3A-dependent DNA methylation is required for spermatogonial stem cells to commit to spermatogenesis. Nature genetics. 2022;54(4):469-80. doi:10.1038/s41588-022-01040-z

14. Na F, Pan X, Chen J, Chen X, Wang M, Chi P, et al. KMT2C deficiency promotes small cell lung cancer metastasis through DNMT3A-mediated epigenetic reprogramming. Nature cancer. 2022;3(6):753-67. doi:10.1038/s43018-022-00361-6

15. Gurevich DB, David DT, Sundararaman A, Patel J. Endothelial Heterogeneity in Development and Wound Healing. Cells. 2021;10(9). doi:10.3390/cells10092338

16. Działo E, Czepiel M, Tkacz K, Siedlar M, Kania G, Błyszczuk P. WNT/β-Catenin Signaling Promotes TGF-β-Mediated Activation of Human Cardiac Fibroblasts by Enhancing IL-11 Production. International journal of molecular sciences. 2021;22(18). doi:10.3390/ijms221810072

17. Venugopal H, Hanna A, Humeres C, Frangogiannis NG. Properties and Functions of Fibroblasts and Myofibroblasts in Myocardial Infarction. Cells. 2022;11(9). doi:10.3390/cells11091386

18. Del Castillo Falconi VM, Torres-Arciga K, Matus-Ortega G, Díaz-Chávez J, Herrera LA. DNA Methyltransferases: From Evolution to Clinical Applications. International journal of molecular sciences. 2022;23(16). doi:10.3390/ijms23168994

19. Blanco-Fernandez B, Castaño O, Mateos-Timoneda M, Engel E, Pérez-Amodio S. Nanotechnology Approaches in Chronic Wound Healing. Advances in wound care. 2021;10(5):234-56. doi:10.1089/wound.2019.1094

20. Wilkinson HN, Hardman MJ. Wound healing: cellular mechanisms and pathological outcomes. Open biology. 2020;10(9):200223. doi:10.1098/rsob.200223

21. Kim IK, Kim K, Lee E, Oh DS, Park CS, Park S, et al. Sox7 promotes high-grade glioma by increasing VEGFR2-mediated vascular abnormality. The Journal of experimental medicine. 2018;215(3):963-83. doi:10.1084/jem.20170123

22. Hennigs JK, Matuszcak C, Trepel M, Körbelin J. Vascular Endothelial Cells: Heterogeneity and Targeting Approaches. Cells. 2021;10(10). doi:10.3390/cells10102712

23. Krüger-Genge A, Blocki A, Franke RP, Jung F. Vascular Endothelial Cell Biology: An Update. International journal of molecular sciences. 2019;20(18). doi:10.3390/ijms20184411

24. Legrand JMD, Martino MM. Growth Factor and Cytokine Delivery Systems for Wound Healing. Cold Spring Harbor perspectives in biology. 2022;14(8). doi:10.1101/cshperspect.a041234

25. Martin RF. Wound Healing. The Surgical clinics of North America. 2020;100(4):ix-xi. doi:10.1016/j.suc.2020.05.012

26. Venugopal K, Feng Y, Shabashvili D, Guryanova OA. Alterations to DNMT3A in Hematologic Malignancies. Cancer research. 2021;81(2):254-63. doi:10.1158/0008-5472.Can-20-3033

27. Norollahi SE, Foumani MG, Pishkhan MK, Shafaghi A, Alipour M, Jamkhaneh VB, et al. DNA Methylation Profiling of MYC, SMAD2/3 and DNMT3A in Colorectal Cancer. Oman medical journal. 2021;36(6):e315. doi:10.5001/omj.2020.93

3.All the figures' quality is so poor, please revise it.

Re: Thank you for your comments. We have improved the resolution of all Figures to 600 dpi.

4.Organize all primer sequences in one table.

Re: Thank you for your valuable suggestion. We have organized all primer sequences in S1 Table as below.

Reviewer #2: The authors demonstrate that DNMT3A-mediated down-regulation of SOX17 promotes endothelial cell migration and TGF-β secretion, which activate fibroblasts and facilitate wound healing. The authors show that SOX17 and DNMT3A have opposite expressions during the process of wound healing. SOX17 knockdown can promote HUVEC cell migration and enhance production of TGF-β. SOX17 silenced-HUVECs-derived conditioned medium co-culture with HFF-1 cells can promote HFF-1 activation. DNMT3A negatively regulated SOX17 expression in HUVECs by increasing its DNA methylation level. There are several issues that I hope the authors could clarify.

Major

1. The in vivo study shows a significant number of new blood vessel are observed on day 7 in wound skin tissue. In vitro study, SOX17 knockdown facilitate HUVEC cell migration (Fig.3D). However, SOX17 level in wound skin tissues is increased from day 1 to day 7 (Fig.2B). The in vivo and in vitro results are reversed. The results are confusing.

Re: Thanks for your professional review work on our article. In fact, the abnormal up-regulation of many proteins during the occurrence of the diseases is regarded as the result of stress in vivo. For example, some tumor suppressor proteins are found to be highly expressed in cancer. CDKN2A gene functions as an important tumor suppressor in various human malignancies, and it prevents carcinogenesis via induction of cell growth arrest and senescence [1-2]. However, based on the data of TCGA database, CDKN2A expression was aberrantly up-regulated in various malignancies, including cervical cancer (CESC), ovarian cancer (OV), sarcoma (SARC), and uterine carcinosarcoma (UCS). The bar chart is downloaded from the GEPIA2 database (http://gepia2.cancer-pku.cn/#general).

The expression of genes in tissues can only indicate that these genes were related to the disease, but it can not directly reflect whether these are positively or negatively correlated with the disease progression. In addition, the expression of SOX17 and DNMT3A showed opposite trends, which is consistent in animal study (Fig. 2B and C) and in vitro experiment (Fig. 5A).

[1] Collado M, Blasco MA, Serrano M. Cellular senescence in cancer and aging. Cell. 2007;130:223–233.

[2] Rayess H, Wang MB, Srivatsan ES. Cellular senescence and tumor suppressor gene p16. Int J Cancer. 2012;130:1715–1725.

2. Transfection of sh-SOX17 for 48h can suppress HUVEC cell viability (Fig.3C) and the authors select 48h transfection for follow-up experiments. Moreover, the following result shows that HUVEC migration number is increased after sh-SOX17 transfection 48h (Fig.3D). How could a decrease in cell viability lead to an increase in migration? The results are puzzling, and the authors need to explain the reasons for the choice of sh-SOX17 transfection time.

Re: Thank you for your careful review of our work. In fact, through CCK8 experiment, we observed that SOX17 silencing for 48 h had a certain inhibitory effect on the viability of HUVECs. However, due to the high requirement of CCK8 itself on seeding cells in 96-well plates and the limitations of the technology itself, this assay has limited accuracy in measuring cell proliferation ability. Therefore, we used the EdU assay to more directly evaluate the effect of SOX17 on HUVEC cell proliferation. We found that SOX17 interference for 48 h could promote cell proliferation to a certain extent, but there was no statistically significant difference (data shown below). The subsequent transwell migration experiment revealed that SOX17 silencing for 48 h markedly promoted HUVEC cell migration, suggesting that the effect of SOX17 on HUVEC cells was mainly to affect migration ability but not cell proliferation. 

3. The authors use the culture medium of cultured HUVEC cells to co-culture with HFF-1 cells. Since ECM medium is used in HUVEC cells culture, while HFF-1 cells use DMEM medium, these two mediums contain different media components, ECM medium contains a variety of growth factors, whether it has an impact on the growth of HFF-1 cells remains to be determined. In addition, HUVEC cell medium suspension contained a variety of HUVEC secreted factors, and it is not clear that TGF-β in the suspension activated HFF-1 fibroblasts. Blocking TGF-β in HUVEC medium suspension and co-culture with HFF-1 can more effectively prove that TGF-β in medium suspension can activate HFF1 fibroblasts.

Re: Thank you for your valuable comments. Although ECM medium contains a variety of growth factors, and whether it has an impact on the growth of HFF-1 cells remains to be determined, HFF-1 cells in different groups were all treated with the culture supernatant of HUVECs. Therefore, the only variable is whether the SOX17 is knocked down.

We agree with you that “Blocking TGF-β in HUVEC medium suspension and co-culture with HFF-1 can more effectively prove that TGF-β in medium suspension can activate HFF1 fibroblasts.”. We were trying conduct rescue experiment to prove that SOX17 interference in HUVECs promoted the activation of HFF-1 through TGF-β. However, TGF-β is a secreted protein, and knockdown assays are not applicable for silencing the expression of TGF-β. We look through a lot of published articles, and drugs used for TGF-β inhibition mainly include LY2157299/galunisertib (Selective TGF-β receptor type I kinase inhibitor), SB431542 (TGF-β receptor kinase inhibitor), LY2109761 (selective inhibitor of TGF-β receptor type I/II) and pirfenidone (pirfenidone inhibited cell growth and reduced TGF-β2 protein levels in human glioma cell lines). Some of the drugs are TGF-β receptor inhibitors with potent activities and may markedly suppress the viability of HFF-1 cells. Therefore, addition of these drugs can not verify the effect of SOX17 knockdown on HFF-1 cells is mediated by TGF-β. In addition, pirfenidone is also a TGF-β inhibitor. However, its working mechanism is not fully understand, and it is not suitable for the rescue experiments.

In our manuscript, we pointed out that “SOX17 interference in HUVECs promoted the activation of HFF-1, and this effect might be mediated by the increased production of TGF-β.”, and whether the effect of SOX17 knockdown on the viability and activation of HFF-1 cells is mediated by TGF-β needs to be further explored. We have added this limitation in the discussion section as below.

There are some shortcomings in this study that should be considered. Whether the effect of SOX17 knockdown on the viability and activation of HFF-1 cells is mediated by TGF-β needs to be further explored. In addition, gene depletion animal model needs to be established to further confirm the conclusion.

4. The immunofluorescence staining of the S100A4, SOX17 and DNMT3A are performed on three different sections (Fig.2), and it is better to stain them on the same section.

Re: Thank you for your kind suggestion. In the preliminary experiment, we tried to perform triple-immunofluorescent staining to analyze the expression of S100A4, SOX17, and DNMT3A on the same section, so that the expression correlation of these three molecules could be better reflected. However, due to the limitations of the technology, the staining effect of triple-immunofluorescent staining was very bad, and then we consider single-IF staining.

5. DNMT3A overexpression reduces the mRNA level of SOX17 in HUVECs, and the authors conclude that DNMT3A negative regulation SOX17 promotes HUVECs migration and wound healing. On day 3 and 7, an increase in the number of new blood vessels is observed, combined with the authors' conclusions, this means the expression of DNMT3A is higher elevated and the expression of SOX17 is decreased on day 3 and 7. However, the in vivo study shows DNMT3A expression is down-regulated with the time extending from day 1 to day 7 and SOX17 expression is opposite of DNMT3A (Fig.2). The authors should explain the these contrary results.

Re: Thank you for your comments. The expression of SOX17 and DNMT3A in vivo appeared to be inconsistent with in vitro experiments. However, the abnormal up-regulation of many proteins during the occurrence of the diseases is regarded as the result of stress in vivo.

In addition, the expression of SOX17 and DNMT3A showed opposite trends, which is consistent in animal study (Fig. 2B and C) and in vitro experiment (Fig. 5A).

Minor

1. There are some inaccuracies in the use of words. For example, in the second paragraph of the discussion section, the authors state that “SOX17 interference induced endothelial cell migration”, among them, “induced” is used inaccurately. In addition, in the third paragraph, “absence” is an inaccurate use in “These data verified that SOX17 absence in HUVECs promoted the viability and activation of HFF-1 cells”.

Re: Thank you for your careful review. We have changed the word “induced” to “promoted” in the sentence “SOX17 interference markedly induced endothelial cell migration.”, and the word “absence” has been replaced by “knockdown” in the sentence “These data verified that SOX17 absence in HUVECs promoted the viability and activation of HFF-1 cells”.

The same errors in the manuscript have been revised, and our manuscript has been polished by a professional language polishing institution.

---

## [Editor Report · Decision Letter 1]

27 Sep 2023

DNMT3A-mediated epigenetic silencing of SOX17 contributes to endothelial cell migration and fibroblast activation in wound healing

PONE-D-23-16896R1

Dear Dr. Zhou,

We’re pleased to inform you that your manuscript has been judged scientifically suitable for publication and will be formally accepted for publication once it meets all outstanding technical requirements.

Kind regards,

Peng Zhang, Ph.D.

Academic Editor

PLOS ONE
---

## [Editor Report · Acceptance letter]

9 Oct 2023

PONE-D-23-16896R1 

DNMT3A-mediated epigenetic silencing of SOX17 contributes to endothelial cell migration and fibroblast activation in wound healing 

Dear Dr. Zhou:

I'm pleased to inform you that your manuscript has been deemed suitable for publication in PLOS ONE. Congratulations! Your manuscript is now with our production department. 

Kind regards, 

on behalf of

Prof. Peng Zhang 

Academic Editor

PLOS ONE